# The Prevalence and Predictors of Restless Legs Syndrome in Patients with Liver Cirrhosis

**DOI:** 10.3390/healthcare10050822

**Published:** 2022-04-28

**Authors:** Oana-Mihaela Plotogea, Camelia Cristina Diaconu, Gina Gheorghe, Madalina Stan-Ilie, Ruxandra Oprita, Vasile Sandru, Nicolae Bacalbasa, Gabriel Constantinescu

**Affiliations:** 1Department 5, “Carol Davila” University of Medicine and Pharmacy, 050474 Bucharest, Romania; plotogea.oana@gmail.com (O.-M.P.); gheorghe_gina2000@yahoo.com (G.G.); drmadalina@gmail.com (M.S.-I.); ruxandraa69@gmail.com (R.O.); gabrielconstantinescu63@gmail.com (G.C.); 2Department of Gastroenterology, Clinical Emergency Hospital of Bucharest, 014461 Bucharest, Romania; drsandruvasile@gmail.com; 3Department of Internal Medicine, Clinical Emergency Hospital of Bucharest, 014461 Bucharest, Romania; 4Department of Visceral Surgery, Center of Excellence in Translational Medicine, Fundeni Clinical Institute, “Carol Davila” University of Medicine and Pharmacy, 022328 Bucharest, Romania; nicolae_bacalbasa@yahoo.ro

**Keywords:** cirrhosis, restless legs syndrome, risk factors, predictors

## Abstract

Introduction and aim. Sleep disorders are highly prevalent in patients with liver cirrhosis. The aim of this study was to investigate the prevalence of restless legs syndrome (RLS), as well as its risk factors and possible predictors, in a cohort of patients with liver cirrhosis. Material and methods. We performed a cross-sectional prospective study over a period of 14 months and enrolled 69 patients with liver cirrhosis, after applying the inclusion and exclusion criteria. The cases of RLS were assessed according to the International Restless Legs Syndrome Study Group (IRLSSG) criteria and severity scale. Results. Out of the total number of patients, 55% fulfilled the criteria for RLS. Age, diabetes, severity of cirrhosis, serum creatinine, glomerular filtration rate (GFR), and mean hemoglobin were associated with the presence of RLS. Moreover, there was a significantly higher prevalence of RLS among patients with decompensated cirrhosis. From all the risk factors introduced into the multivariate analysis, only the GFR could predict the presence of RLS. Conclusions. This research shows that patients with liver cirrhosis have a high risk of RLS. Even though there are multiple risk factors associated with RLS, only the GFR could predict its occurrence in our cohort.

## 1. Introduction

Liver cirrhosis is an important cause of mortality and morbidity worldwide. Patients with cirrhosis may evolve from asymptomatic or compensated phases to symptomatic or decompensated phases, with complications such as variceal bleeding, hepatic encephalopathy, hepatorenal syndrome, and others. Most cirrhosis complications require multiple hospitalizations and result in high rates of mortality and compromised health-related quality of life (HRQOL) [1]. Apart from life-threatening complications, several other conditions contribute to HRQOL impairment. A shift in this paradigm has recently been embraced, with more emphasis given to the extrahepatic conditions that compromise HRQOL, particularly sleep disorders and restless legs syndrome (RLS) [2,3,4,5].

RLS, also known as Ekbom’s syndrome, is a sleep-related movement disorder characterized by sensory and motor neurological symptoms with a circadian pattern, manifested as the urge to move, primarily the legs and other extremities, during rest [6,7]. These movements are also a relief for unpleasant somatosensory sensations that worsen during the evening and night [7]. RLS has been described as a primary (idiopathic) condition, with a prevalence of 2.5–15% among the general population [8,9], and as a secondary condition, associated with different pathologies, with higher prevalence rates: chronic kidney disease (up to 68%), iron deficiency anemia, uremia, neuropathy, idiopathic pulmonary fibrosis, irritable bowel syndrome, inflammatory bowel disease, rheumatoid arthritis, and others [10,11,12].

The prevalence of RLS in patients with chronic liver disease was first assessed by Franco et al. in 2008 and reported as 62% [9]. Meta-analyses showed that the prevalence of RLS is variable among different populations of patients with cirrhosis worldwide, depending on the presence of associated conditions, the diagnostic criteria, and the questionnaires used. However, despite all the differences between studies, the prevalence of RLS in patients with chronic liver disease is higher than in the general population [5].

The role of liver cirrhosis in RLS and the pathophysiological patterns underlying it are controversial. Various conditions are encountered in cirrhosis (such as iron deficiency, anemia, and neuropathy) that are also considered risk factors for RLS occurrence. The predictors and mechanisms of RLS in cirrhotic patients are incompletely elucidated and the direct association between RLS and cirrhosis is not clear. However, the major negative impact of both cirrhosis and RLS on HRQOL is indisputable [9].

The main objective of this study was to investigate the prevalence of RLS in patients with compensated and decompensated cirrhosis from an emergency hospital in Romania. The secondary objectives were to identify the potential contributing risk factors and predictors of RLS in patients with cirrhosis. The strength of the present study consists in evaluating a poorly understood and underestimated extrahepatic condition encountered among patients with cirrhosis. Taking into consideration that RLS has never before been studied among Romanian cirrhotics, we considered it necessary to investigate the prevalence and predictors of RLS in order to ensure early detection and even treatment, so as to provide better patient outcomes and HRQOL.

## 2. Materials and Methods

### 2.1. Patients

We conducted a prospective cross-sectional study in the Clinical Emergency Hospital of Bucharest, Romania, between December 2020 and January 2022 (14 months).

In total, 94 adults were initially considered for enrolment, based on the diagnosis of compensated or decompensated cirrhosis in accordance with clinical, laboratory, ultrasound, and endoscopy findings. The inclusion criteria referred to either hospitalized patients or patients presenting for regular follow-ups. The exclusion criteria included overt hepatic encephalopathy (HE) (clinically defined as grades 2–4, according to West Haven criteria), severe renal dysfunction (creatinine > 2 mg/dL), unstable cardiovascular status, acute or chronic neurologic conditions, non-hepatic and/or hepatic malignancies. All data were obtained one day before or on the day of discharge. Patients hospitalized for variceal bleeding were not excluded from the study, but were enrolled on the day of discharge, given that a low hemoglobin value at admission would have added bias to the statistical analysis. Patients admitted for hepatorenal syndrome were excluded if their creatinine values did not decrease below 2 mg/dL.

After applying the exclusion criteria, 69 patients were ultimately included in the study.

We collected patients’ data, including gender, age, smoking status, comorbidities (diabetes), cirrhosis-related parameters (etiology, severity according to Child–Pugh score), paraclinical data (hemoglobin, serum creatinine, and glomerular filtration rate (GFR)). GFR was calculated using MDRD (Modification of Diet in Renal Disease) GFR Equation [13].

### 2.2. RLS Assessment

First, we applied the 2014 International Restless Legs Syndrome Study Group (IRLSSG) criteria [14]:

1. An urge to move the legs, usually accompanied or caused by uncomfortable and unpleasant sensations in the legs.

2. The urge to move or the beginning or worsening of unpleasant sensations during periods of rest or inactivity, such as lying or sitting.

3. The urge to move or unpleasant sensations that are partially or completely relieved by movement, such as walking or stretching, at least as long as the activity continues.

4. The urge to move or unpleasant sensations that are worse in the evening or night than during the day or occur only in the evening or night.

Secondly, we administered the IRLSSG severity scale [15], translated in Romanian and provided by Mapi Research Trust [16], a questionnaire consisting of 10 questions with scores ranging from 0 to 40. Depending on the sum of the points, RLS was classified as mild (0–10 points), moderate (11–20 points), severe (21–30 points), or very severe (31–40 points) [15].

### 2.3. Ethical Aspects

The study was approved by the Research Ethics Committee of the Clinical Emergency Hospital of Bucharest, Romania (approval no. 3928/11.02.2020). Patients’ data were anonymously noted from their medical records, while the completion of the questionnaires was considered the implied consent to participate. The study was conducted according to the Declaration of Helsinki (1975), as revised in 2008, for medical research involving human subjects [17].

### 2.4. Statistical Analysis

Data were collected in Microsoft Excel and statistically analyzed with IBM SPSS v. 20 software package program. Descriptive analysis was performed for the prevalence of RLS in the study group and comparison of demographic, clinical, and paraclinical variables between patients with RLS and patients without RLS. Continuous variables were expressed as mean ± standard deviations and ranges or as medians and ranges. Categorical variables were expressed as frequencies/absolute numbers with percentages. Subgroup differences were tested using either chi-square test or unifactorial ANOVA. For multivariate analysis, we used logistic regression with standard method (also known as enter method) by introducing all the independent variables in the equation simultaneously. Any *p*-values <0.05 were indicative of statistical significance.

## 3. Results

### 3.1. Patients’ Demographic and Paraclinical Data

In total, 69 patients with cirrhosis were included during the study period, with a mean age of 63.17 ± 7.78 years and a gender distribution of 41 males and 28 females. Thirty-eight patients fulfilled the four criteria for the diagnosis of RLS, indicating a prevalence of 55%. We further compared the demographic and paraclinical data between the two subgroups (non-RLS vs. RLS) (Table 1).

There was a statistical difference regarding age, as the patients with RLS were older than those without RLS. There was no difference in smoking status between the RLS patients and the non-RLS patients (*p* = 0.052). Diabetes was more frequently encountered among the patients with RLS compared to the non-RLS patients (52.6% vs. 19.4%, *p* = 0.004). The serum creatinine levels were higher in the patients with RLS and, therefore, the GFR was significantly lower in the patients with RLS, who were also older (taking into consideration that age is a parameter for GFR calculation).

Overall, anemia was present in 56.5% of the patients with cirrhosis, with a mean hemoglobin value of 11.61 ± 1.53 mg/dL. The patients with RLS had a mean haemoglobin value of 10.89 ± 1.41 md/dL, which was significantly lower than that of the patients without RLS: 12.50 ± 1.19 mg/dL (*p* < 0.001). The most frequently encountered type of anemia was normocytic.

In total, 50% of the patients with RLS had severe and very severe symptoms, according to the IRLSSG severity scale.

We further analyzed the etiology, type (compensated or decompensated) and severity (Child–Pugh score) of cirrhosis by comparing the patients with RLS and the patients without RLS (Table 2). Regarding the etiology, there were no statistical differences between the non-RLS patients and the RLS patients (*p* = 0.109). The most frequent causes of cirrhosis in the patients with RLS were related to alcohol consumption (11 patients had alcoholic cirrhosis and 15 patients had mixed-alcoholic-and-viral cirrhosis).

Concerning the type of cirrhosis, which was either compensated or decompensated, the results showed that significantly more patients with RLS had decompensated cirrhosis compared to the non-RLS patients (76.35% vs. 38.7%, *p* = 0.002). Consequently, the Child–Pugh score was significantly higher among the patients with RLS compared to the non-RLS patients (*p* = 0.001).

### 3.2. Predictors of RLS

We also investigated the predictors of RLS in our study group, by selecting the parameters that proved to statistically differentiate the patients without RLS from those with RLS. Consequently, by introducing the variables into the multiple regression analysis, out of all the risk factors associated with RLS, only the GFR could predict the presence of RLS (Table 3), with an accuracy of 79.7%. Moreover, the ROC curve was used to obtain the sensitivity and specificity of the GFR value, to determine whether the patients with cirrhosis also had RLS. The ROC curve revealed a specificity of 83% and a sensitivity of 81%, and the area under the curve (AUC) was reliable, confirming the validity of the test as 89% (Figure 1).

## 4. Discussion

Cirrhosis is the terminal stage of chronic liver disease and may clinically evolve from asymptomatic to decompensated phases. The consequences attributed to liver dysfunction are very numerous, ranging from portal hypertension and its manifestations (variceal bleeding, hepatic encephalopathy, ascites, and others) to a variety of extrahepatic complications, such as cardiac, renal, pulmonary, neurological, and other conditions [18].

RLS is one of the most common sleep disturbances encountered in patients with liver cirrhosis, with a multifactorial etiology that is not completely elucidated. It is not clear whether RLS appears as a direct consequence of impaired liver function, or whether it is determined by associated pathologies. Apart from genetic and environmental factors, which have been well documented in the pathophysiology of primary (idiopathic) RLS, several studies have suggested various specific mechanisms for RLS occurrence in patients with cirrhosis, such as decreased serotoninergic and dopaminergic transmissions, increased gamma-aminobutyric acid (GABA) and glutamate transmissions, corticospinal tract dysfunction, elevated toxins and cytokines, metabolic disorders (cerebral iron deficiency, hypokalemia, hyponatremia, vitamin B12, and folate deficiency) [5,9,19,20,21,22]. Recently, the involvement of chronic inflammation has also been proposed as being responsible for the occurrence of RLS. High levels of serum C-reactive protein were associated, independently of other variables (e.g., gender, age, body mass index, and smoking status), with RLS symptoms. Moreover, subjects with increased interferon (IFN)-α autoantibodies were four times more likely to experience RLS than those with low levels [23]. In addition to systemic inflammation, there are data in the literature that suggest an important role of hypoxia in the pathophysiology of RLS. Hypoxia-inducible factors (HIFs), such as HIF-1α, are high in patients with RLS [24]. As a particularity, HIF-1α levels are also increased in the brains and livers of patients with NAFLD and in those with obstructive sleep apnea, aggravating the progression of liver disease and chronic intermittent hypoxia [25,26]. Therefore, these findings suggest that hypoxia pathways are multifactorial and that they are additionally implicated in other mechanisms in the appearance of RLS among patients with liver diseases.

The recognition of RLS is very important in clinical practice, since it has a tremendous effect on quality of life. Several studies worldwide have assessed the prevalence of RLS in cirrhotic patients [20,21,22,27]. To our knowledge, the present study is the first research in Romania investigating the presence of RLS among these subjects. In our study, we found a prevalence of RLS of 55% RLS among a cohort of 69 cirrhotic patients. A higher prevalence (62%) was reported in the USA by Franco et al. [9], while the lowest prevalence (16.8%) was reported among Japanese cirrhotic patients [21]. These wide ranges of prevalence rates indicate, on one hand, that ethnicity and genetics may play a role in the occurrence of RLS, and, on the other hand, that the heterogenicity of the assessment methods influences the results. The main screening tool for the diagnosis and severity rating of RLS was developed by the International Restless Legs Syndrome Study Group [14,15]. These criteria are used among both healthy controls and study groups with different pathologies. However, in the literature, there are various other diagnostic methods, from clinical interviews and examination to telephone interviews or questionnaires, such as the Cambridge-Hopkins Questionnaire for the Assessment of Restless Legs Syndrome or the Johns Hopkins Telephone Diagnostic Interview for RLS [5,20,27].

Regarding the risk factors and predictors of RLS, there are contradictory data. In our study, age, diabetes, severity of cirrhosis, serum creatinine, GFR, and mean hemoglobin were associated with the presence of RLS. The prevalence of RLS increased proportionally with age and was higher in patients with anemia, high creatinine, and low RFG values. Moreover, there was a significantly higher prevalence of RLS among the patients with decompensated cirrhosis. Similarly, Rajender et al. also reported strong associations between the severity of liver cirrhosis and the presence of RLS [8]. Unlike these findings, other researchers reported that the severity of cirrhosis measured by Child–Pugh score did not influence the prevalence of RLS [21,22,28]. In our opinion, these differences could be attributable to confounding factors that were not excluded from the analysis, such as cirrhosis’ complications (hepatorenal syndrome, gastrointestinal bleeding, and minimal HE). Nevertheless, there seemed to be a close relationship between several risk factors and the presence of RLS. When applying the multivariate analysis, we observed that only GFR could predict the occurrence of RLS in our study group.

Concerning the etiology, we did not find any relationship between the cause of cirrhosis and the presence of RLS, which was in line with the findings of other researchers [20,21,22]. One study reported that patients with alcohol cirrhosis exhibited RLS in significantly higher percentages, unlike other etiologies, in which the distribution of RLS was homogeneous [8]. Researchers have also reported symptoms of RLS in patients with chronic hepatitis C as a complication of IFN-α treatment [29]. By contrast, treatment with direct-acting antivirals decreased the prevalence and severity of RLS symptoms among patients with chronic hepatitis C [30].

RLS may also occur in combination with other sleep disorders. Interestingly, in a cohort of 42 patients with primary biliary cirrhosis, researchers found that a high number of subjects reported RLS associated with poor sleep quality and daytime sleepiness, measured by both subjective and objective methods [31].

It is well-known that patients with cirrhosis have impaired HRQOL compared to healthy controls, mainly because of cirrhosis’ complications. However, there is increasing evidence that even in the compensated stage of cirrhosis, patients record significantly low scores when completing HRQOL assessment questionnaires. The presence of RLS in cirrhotic patients has been reported to induce anxiety, daytime fatigability, depression, and cognitive impairments [5]. Furthermore, RLS has economic consequences, including reduced work productivity [19]. These findings are attributed to associated conditions, such as the subclinical cognitive impairment of HE or non-obvious manifestations of sleep disturbances [3,4,32,33].

Given the high prevalence of RLS and the great emphasis that has recently been placed on atients’ HRQOL, the present study increases the awareness of clinicians for diagnosing and treating patients with cirrhosis and RLS. treatment is generally recommended for moderate and severe symptoms. Pharmacological treatment consists of α2δ ligands (gabapentin enacarbil, pregabalin), dopaminergic agents (ropinirole, cabergoline etc.), or opioids (oxycodone/naloxone) [34]. However, the majority of these drugs should be avoided in cirrhosis, due to the associated risk of precipitating hepatic encephalopathy. Nonpharmacological therapies include dietary protein restriction, massage, walking, temperate warm or cold baths, and regular sleep–wake cycles [35]. Moreover, it is very important to identify and treat factors that could aggravate RLS in patients with cirrhosis, such as iron deficiency, metabolic disorders, and renal insufficiency. Further research in this area is needed to clarify the complex relationship between cirrhosis and RLS and, more importantly, to guide the clinical management of these patients.

This study features several limitations. First, we studied the prevalence of RLS among subjects with a presumed pathology, given that all the patients enrolled had cirrhosis, which meant that we lacked a healthy control cohort. Secondly, this study was performed according to subjective manifestations and self-reported questionnaires, which are prone to bias. Ultimately, even though the aim was not to investigate HRQOL in relation to RLS, we admit that this would have added value to our research, and we aim to focus on this topic in future studies.

## 5. Conclusions

This research showed that patients with liver cirrhosis experience high rates of RLS. Even though there are multiple risk factors associated with RLS, only GFR could predict its occurrence in our cohort.

The impact of RLS is important in various areas of daily life. Therefore, more large, prospective studies are necessary to assess the prevalence of RLS, as well as potential risk factors and treatment options that could improve patients’ symptoms and HRQOL.

## Figures and Tables

**Figure 1 healthcare-10-00822-f001:**
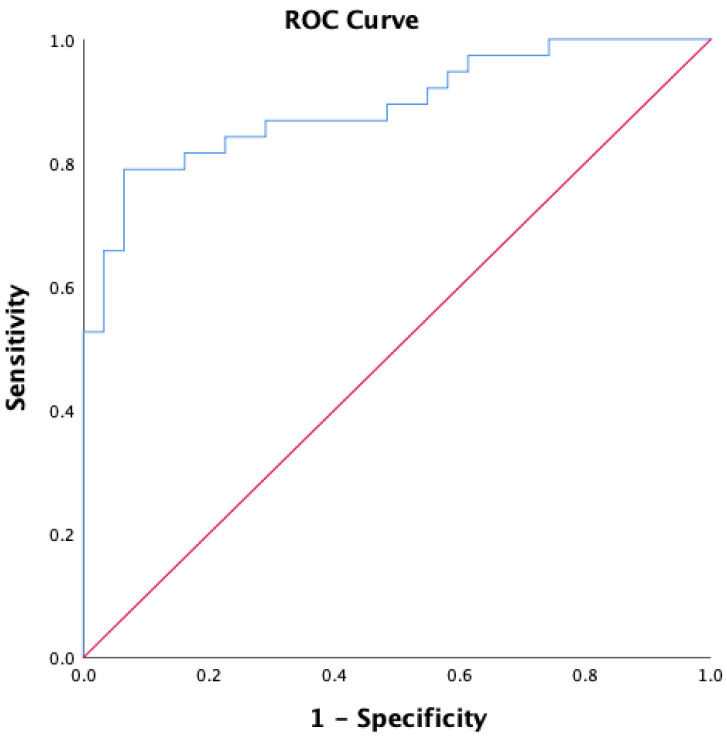
ROC curve for predictors of RLS (AUC = 0.891; sensitivity = 0.816; specificity = 0.839).

**Table 1 healthcare-10-00822-t001:** Comparisons between the two subgroups (patients with RLS and patients without RLS) regarding demographic and paraclinical data.

Demographic and Paraclinical Parameters	All Patients (*n* = 69)	Non-RLS (*n* = 31, 44.92%)	RLS (*n* = 38, 55.07%)	*p*-Value
Age (years), mean ± SD	63.17 ± 7.78	59.94 ± 7.09	65.82 ± 7.39	0.001 *
Gender (M/F), *n* (%)	41/28 (59.4/40.6%)	20/11 (64.5/35.5%)	21/17 (55.3/44.7%)	0.298
Smoking, (Yes), *n* (%)	23 (33/3%)	14 (45.2%)	9 (23.7%)	0.052
Diabetes, (Yes), *n* (%)	26 (37.7%)	6 (19.4%)	20 (52.6%)	0.004 *
Creatinine (mg/dL), mean ± SD	1.21 ± 0.31	1.08 ± 0.21	1.32 ± 0.34	0.001 *
GFR (mL/min/1.73 m^2^), mean ± SD	61.69 ± 19.25	72.32 ± 17.19	53.01 ± 16.41	<0.001 *
GFR groups				
Normal (≥90)	3 (4.3%)	3 (9.7%)	0 (0%)	0.001 *
Mild decrease (60–89)	31 (44.9%)	20 (64.5%)	11 (28.9%)	
Moderate decrease (30–59)	33 (47.8%)	8 (25.8%)	25 (65.8%)	
Severe decrease (<30)	2 (2.9%)	0 (0%)	2 (5.3%)	
Haemoglobin (g/dL), mean ± SD	11.61 ± 1.53	12.50 ± 1.19	10.89 ± 1.41	<0.001 *
Type of anemia, *n* (%)				
No anemia	30 (43.5%)	20 (64.5%)	10 (26.3%)	0.001 *
Macrocytic anemia	10 (14.5%)	6 (19.4%)	4 (10.5%)	
Microcytic anemia	9 (13%)	1 (3.2%)	8 (21.1%)	
Normocytic anemia	20 (29%)	4 (12.9%)	16 (42.1%)	
RLS Score, mean ± SD	10.90 ± 12.34	-	19.79 ± 9.96	-
RLS severity, *n* (%)				
No RLS	31 (44.9%)	31 (100%)	-	-
Mild	7 (10.1%)	-	7 (18.4%)	
Moderate	12 (17.4%)	-	12 (31.6%)	
Severe	9 (13%)	-	9 (23.7%)	
Very severe	10 (14.5%)	-	10 (26.3%)	

Legend: GFR = glomerular filtration rate; M = male; F = female; SD = standard deviation; GFR = glomerular filtration rate; * *p* < 0.05.

**Table 2 healthcare-10-00822-t002:** Comparisons between patients with RLS and patients without RLS regarding etiology, type, and severity of liver cirrhosis.

Cirrhosis-Related Parameters	All Patients (*n* = 69)	Non-RLS (*n* = 31)	RLS (*n* = 38)	*p*-Value
Cirrhosis etiology, *n* (%)				
Alcoholic	19 (27.5%)	8 (25.8%)	11 (28.9%)	0.109
Viral hepatitis	23 (33.3%)	14 (45.2%)	9 (23.7%)	
Alcoholic + viral hepatitis	20 (29%)	5 (16.1%)	15 (39.5%)	
NAFLD	7 (10.1%)	4 (12.9%)	3 (7.9%)	
Cirrhosis type, *n* (%)				
Compensated	28 (40.6%)	19 (61.3%)	9 (23.7%)	0.002 *
Decompensated	41 (59.4%)	12 (38.7%)	29 (76.35%)	
Cirrhosis severity according to Child–Pugh score, *n* (%)				
Child A	28 (40.6%)	19 (61.3%)	9 (23.7%)	0.001 *
Child B	17 (24.6%)	8 (25.8%)	9 (23.7%)	
Child C	24 (34.8%)	4 (12.9%)	20 (52.6%)	

Legend: NAFLD = non-alcoholic fatty liver disease; * *p* < 0.05.

**Table 3 healthcare-10-00822-t003:** Logistic regression analysis for predictors of RLS.

	Multiple Regression			
Variables	OR [95% CI]	Coefficient Beta	*p*-Value	Predicted Percentage
Age	0.98 [0.87–1.10]	−0.01	0.79	79.7%
Creatinine	0.00 [0–2.46]	−5.67	0.09	
GFR	0.89 [0.79–0.99]	−0.11	0.04 *	
Type of anemia				
No anemia	REF			
Macrocytic anemia	0.10 [0.00–1.98]	−2.30	0.13	
Microcytic anemia	0.74 [0.01–42.93]	−0.29	0.88	
Normocytic anemia	0.60 [ 0.03–11.70]	−0.49	0.71	
Hemoglobin	0.31 [0.09–1.08]	−1.15	0.06	
Diabetes	2.73 [056–13.31]	1.00	0.21	
Cirrhosis type	1.60 [0.20–12.66]	0.47	0.65	
Cirrhosis severity	0.81 [0.06–9.56]	−0.20	0.87	

Legend: GFR = glomerular filtration rate (mL/min/1.73 m^2^); * *p* < 0.05.

## Data Availability

Data from the study are available upon request from the first author (O.-M.P.)

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
