# Peer review of "The Prevalence and Predictors of Restless Legs Syndrome in Patients with Liver Cirrhosis"

_healthcare, 2022, doi:10.3390/healthcare10050822_

Round 1

Reviewer 1 Report

This study investigated the prevalence of RLS, risk factors, and possible predictors in participants with liver cirrhosis. I raise several concerns.

  1. Authors should further elaborate on the need for and novelty of the study in the introduction section. What are the strengths of this study over other similar studies? Why is this study needed?
  2. The number of study subjects is very small. Is the normal distribution satisfactory? Was the research method properly reflected in accordance with the data distribution? ANOVA and logistic regression analysis must be used to suit the distribution.
  3. Are all independent variables adjusted for multivariable analysis (Table 3)?
  4. There is no mention of the ROC curve throughout the study, is it necessary to add it as a research result?
  5. Does this study have any limitations? Authors should elaborate on the limitations of the study in detail.

Reviewer 2 Report

The manuscript is informative and well organized. Please find the following comments: 

1- The manuscript needs to be reviewed by English-editor, and to be more organized and consistence

2- The discussion is poor as not recommend clinical nor research

3- Please clarify the role of liver cirrhosis in restless leg syndrome in the introduction and discussion 

Round 2

Reviewer 1 Report

  1. Please detail the methods and results used by the authors to review the normal distribution assumptions in the Statistics section.
  2. Please describe in more detail the key limitations of your study. (e.g. small sample, cross-sectional design, causal inference)
